# Reliability Associated with the Measurement of Continuous Variables in Veterinary Medicine: What the Different Possible Indicators Tell, and How to Use and Report Them

**DOI:** 10.3390/ani13172793

**Published:** 2023-09-02

**Authors:** Sébastien Buczinski

**Affiliations:** Département des Sciences Cliniques, Faculté de Médecine Vétérinaire, Université de Montréal, St-Hyacinthe, QC J2S 2M2, Canada; s.buczinski@umontreal.ca; Tel.: +1-450-773-8521 (ext. 8675)

**Keywords:** intra-class correlation coefficient (ICC), Passing–Bablok regression, Deming regression, Lin’s concordance correlation coefficient

## Abstract

**Simple Summary:**

Veterinary science is based on data collection at the animal or herd level. Beyond the variability in the variable in question, the data collected can depend on the device used or the person performing the measurement. Determination of these sources of variation is crucial to be able to use these measurements in practice or research. In this manuscript, I review the multiple indicators that can be used for determining these sources of variability in order to obtain robust indicators that are useful when trying to quantify test–retest reliability (between multiple measurements by different devices or operators). I also present the pros and cons of each indicator in the absence of a “one size fits all” framework to report them adequately depending on the specific context.

**Abstract:**

Reliable indicators of health status (heart rate, rectal temperature, blood marker, etc.) are of cornerstone importance in the daily practice of veterinary medicine. The reliability of a measurement assesses the variability that is associated with the variable to be measured itself vs. other sources of variation (measurement device, person performing the measurement, etc.). Quantitative and continuous indicators are numerous in practice and the determination of their reliability is a complex issue. In the absence of a gold standard approach, several indicators of reliability have been described and can be used depending on several assumptions, study design, and type of measurement. The aim of this manuscript is, therefore, to determine the applicability of commonly described reliability indicators. After a description of the different sources of errors of a measurement, a review of the different indicators that are commonly used in the veterinary field as well as their applicability, limitations, and interpretations is performed.

## 1. Introduction

Veterinary medicine, as with many scientific fields, is based on many observations associated with the measurement of various physical, clinical, and paraclinical parameters. The measurement of numerical variables is, therefore, a daily task in veterinary science. Measuring rectal temperature, animal weight, or a specific blood marker are common tasks for both clinical and research fields. The objective of any measurement is to determine the biological variability in the variable under interest, in order to take an action based on its results. One of the specific challenges is, therefore, to know how the measurement that is taken is representative of the “true” value of the veterinary patient. In other words, we need to know if the measure obtained is really representative of the patient’s characteristics vs. other sources of variation. If the outcome measured is unreliable, practical consequences will be that the measurement and its change would not be representative of the “true” patient changes. The associated “noise”, due to unreliability, would exceed the “signal” (=true variable change) that needs to be captured to take adequate action. In a research setting, another consequence is that studies focusing on this measurement will be associated with a reduction in the study power via increasing the variance of the outcome [1]. Concepts of reproducibility (two consecutive measurements of the same marker give similar results) and reliability (how measurements from the same veterinary patient can be distinguished from the other despite variable sources of measurement error) are very closely related topics [2]. For these reasons, it is important to assess measurement reliability before using it as an outcome or as a covariate in any specific study. Quantifying the sources of variability in a measurement is of primary importance to judge whether it can be suitable for being used in research and in practice; however, there are multiple ways to assess these characteristics. The multiplicity of these tools may add confusion for the researchers trying to assess the reliability of a quantitative measure. The objective of this review manuscript is, therefore, to outline the important considerations on reliability before applying or using a new measurement tool or device. A test or measurement is said to be reliable when it gives the same result in a patient or sample if measured repeatedly using the same or different devices or operators/technicians [2].

## 2. Key Concepts for Distinguishing Variability in the Numerical Variable to Be Measured and Other Sources of Variability

When trying to assess a specific variable (let us say *M*, that, for example, is the true rectal temperature of a patient, or the true heart rate of a patient), we obtain a “picture” of this variable using our measurement device (a specific value, *M_m_*, which is obtained from a specific thermometer, or the manual counting of the heart beats using a stethoscope by a specific operator). The measurement *M_m_* of the variable *M* quantifies the true value of *M* (which is not known in most cases) plus a specific error term (*ε*). This can also be more formally written as follows (Equation (1)):(1)$$M_{m}=M+\varepsilon$$

This previous equation takes into account the fact that the measurement *M_m_* is just a specific way to assess the true variable of interest (*M*). In the classical measurement theory, the error term *ε* is supposed to be independent of *M* and normally distributed around 0 with a variance $\sigma_{\varepsilon }^{2}$ [2]. The *M* and *M_m_* values are fixed for any individual at a specific moment; however, in a specific population where the same measurements are performed, the independence between *M* and *ε* can be translated in terms of variances.
(2)$$\sigma_{M_{m\;}}^{2}=\sigma_{M}^{2}+\sigma_{\varepsilon }^{2}$$

Equation (2), therefore, shows that the variability in the measurement taken in a specific population has two components that are related to the real difference between veterinary patients and the random error [1]. The measurement is clinically useful if the variance of the error ($\sigma_{\varepsilon }^{2}$) is small enough vs. the variance of the specific variable to assess ($\sigma_{M}^{2}$). This can be more formally written in general terms of reliability in Equation (3):(3)$$Reliability=\frac{\sigma_{M}^{2}}{\sigma_{M_{m}}^{2}}=\frac{\sigma_{M}^{2}}{\sigma_{M\;}^{2}+\sigma_{\varepsilon }^{2}}$$

It can be easily understood that the more reliable the measurement is, the highest part of the variability observed is due to the true variable to assess (*M*) vs. all other sources of error (ε), which tend to 0. This is the simplest model in classical measurement theory [2]. The generalizability theory partitions the variance of error ($\sigma_{\varepsilon }^{2}$) in different error types that can be observed with the different (1, …, l) sources of variation ($\sigma_{\varepsilon_{1}}^{2}$,…, $\sigma_{\varepsilon_{l}}^{2}$, and the residual term $\sigma_{\varepsilon_{r}}^{2})$.

These general concepts also indicate that reliability lies between 0 and 1 as a ratio of variance (which is positive). The more reliable the technique, the closer to 1 the reliability is. In this case, most of the measurement variability is coming from item M and not from the random error term. When comparing two or more ways to assess the same characteristic (let us say *M*_1_, *M*_2_, …) applied to the same population (the same test performed by different raters, e.g., different veterinarians estimating rectal temperature with the same thermometer; testing different thermometer models; or testing different raters testing different thermometers), three general conditions may be observed as defined in classical measurement theory:
The tests can be said to be parallel if their means are equal (μ_1_ = μ_2_) as well as the variance of their errors $\sigma_{\varepsilon 1}^{2}$ = $\sigma_{\varepsilon 2}^{2}$. This implies that the measurements that are parallel obtained by the two different measurements or techniques are interchangeable. For a specific patient, the values of the two different measurements only differ based on the magnitude of the variance of error $\sigma_{\varepsilon }^{2}$. This definition also implies that since the variance of the two tests is equal, their correlation with a third variable should also be equal;The tests may only differ from a specific constant C: *M_m_*_1_ = *M_m_*_2_ + α. In this case, they are called “essentially tau dependent”. They are called Tau-dependent in the special case of α = 0. In all these cases, the variance is not assumed to be constant as for parallel tests. The denomination Tau comes from the way the “*M_m_*” has been historically written as the Greek letter “*τ*” in classical test theory;The last scenario is when the two tests are linearly dependent, which can be written as *M_m_*_1_ = *β* × M*_m_*_2_ + α, where *β* is any real number. This situation defines congeneric tests.


As initially mentioned, it is difficult to know a priori what the specific conditions that we are facing in practice are. Different strategies can be used for validating one of these three situations. I will not go through a detailed assessment of these three different definitions in the current manuscript. The reader is referred to specific references on this topic [3,4]. These different scenarios have been initially created based on psychometric scales, which were developed to determine constructs that cannot be easily measured with one specific instrument (e.g., measuring sociability or anxiety in a particular study using various scoring scales). The specific ε term needs to be further decomposed in terms of error of measurement device, error due to operator, or any other remaining cause of error, as developed in the next section.

When trying to assess the differences observed between two different measurement methods of the same parameter or between two operators/technicians assessing a specific variable with a numeric parameter, different approaches can be taken [2]. Some discrepancy is expected due to either random error and/or specific bias. The main issue is, therefore, to determine to what extent these mechanisms occur due to the variability in the measurement being taken, in order to correctly interpret the results. In order to illustrate this in the manuscript, I have used an open-access dataset used for reporting the reliability of two different veterinarians (operator 1 and operator 2) for the assessment of the maximal depth of ultrasonographic lung consolidation assessments (in cm) of 50 video loops from feedlot calves with or without respiratory problems [5]. The data used, as well as the specific information to reproduce the figures and obtain reliability indicators, are included as Appendix A. A small positive random error (mean = 0, sd = 0.2 cm) has also been added, to avoid data points overlapping.

## 3. Correlation between Two Quantitative Variables (Pearson’s or Spearman’s Correlation Coefficients) Is Not a Reliability Indicator

Establishing the correlation between two quantitative variables is a commonly performed analysis with either Pearson’s or Spearman’s correlation coefficient determination. Pearson’s correlation coefficient (R) can first be seen as an intuitive and natural way to determine a correlation between two variables (Figure 1).

These coefficients are assessing correlations, which are different from the reliability. Pearson’s R coefficient assesses the strength of the linear correlation between these two variables. Pearson’s R, for measuring the association of two different variables *M*_1_ and *M*_2_ (n different pairs of measurements), is written as follows in Equation (4):(4)$$R=\frac{n(\sum M_{1}\times M_{2})-(\sum M_{1})(\sum M_{2})}{\sqrt{\left[n\sum {M_{1}}^{2}-{\left(\sum M_{1}\right)}^{2}\right]}\times \sqrt{\left[n\sum {M_{2}}^{2}-{\left(\sum M_{2}\right)}^{2}\right]}}$$

Therefore, it can roughly be understood as the ratio of covariance between the variables to the product of these variables’ standard deviation. The R-squared value (R^2^) corresponds to the total proportion of variance of the dependent variable (on the *Y*-axis), which can be explained by the linear regression of the dependent variable on the independent variable (on the *X*-axis). This correlation is different from the reliability since highly correlated measures do not mean that these two measurements are interchangeable [6]. Pearson’s R is insensitive to the absolute magnitude of the two methods’ differences. In cases of essentially Tau-dependent and congeneric tests, despite a perfect Pearson’s R value (R = 1), the two tests cannot automatically be used interchangeably. This illustrates why it is not a reliability indicator.

Moreover, Pearson’s R evaluation also depends on the bivariate data distribution and can be heavily influenced by outliers. Spearman’s rho (r) coefficient is a rank-order coefficient that is robust to any distribution of the two variables being compared. Spearman’s r assesses the direction and the strength of direction between the two ranked variables (Figure 1). The variable values, by themselves, are not used for the calculation but rather the ranked variable, which makes Spearman’s r more robust, especially for bivariate data that clearly deviates from normality. Interpretation of Spearman’s r is a little different from Pearson’s R. The higher its value, the higher the correlation is between the ranked variables. It is, therefore, easy to understand that it cannot be interpreted as a way to assess if the two measurements are interchangeable. Several benchmarks for interpreting these coefficients in terms of importance of the correlation have been reported; for example, a negligible (0–0.10), weak (0.10–0.39), moderate (0.40–0.69), strong (0.70–0.89), and very strong (0.90–1.00) positive correlation [6]. However, it is important to remember that these benchmarks are arbitrarily defined and not specifically validated. The take-home message from this section is that the correlation is not equivalent to reliability and that the calculation of a specific correlation measurement is not enough to state the exchangeability between these variables.

## 4. Is There a Difference between Two Measurement Methods, or Two or Plus Different Raters When Performing Ordinal Measurements?

Beyond the limitations of correlation coefficients, another limitation of the previous approaches is that comparisons are mostly limited to paired comparisons (two technicians using the same instrument/technique or two instruments/techniques used by the same technician or device) and cannot be extended where >2 technicians or instruments/techniques are to be compared; however, reliability studies are generally trying to assess 1,2,…,k raters and or instruments/techniques. Intra-class correlation (ICC) coefficients have, therefore, been developed to address this particular need [7]. The ICC coefficients are simply extending Equation (3), where the error (*ε*) is partitioned in the different sources of the variance depending on the study design. This calculation also comes with strong assumptions that data are normally distributed and variances between the measurements (*M_m_*_1_, *M_m_*_2_, …, *M_mk_*) are homoscedastic. The ICC coefficients have also been employed extensively for comparing different scoring scales used by different raters in the psychology field. Despite the fact that the scores cannot be considered continuous variables, they generally meet ICC assumptions. The choice of which particular ICC coefficient to choose is a complex but important debated topic that has been recently reviewed [8]. Most available statistical software allows for the calculation of various ICC coefficients and it is important to choose the reported ICC coefficient correctly and not based on the best obtained value [8].

There are two major observational study designs for these reliability studies, with either (1) a one-way design, where one rater makes multiple measurements of different patients, or (2) a two-way design, where multiple raters obtain measurements of each patient. All raters can assess all patients or some pairs of raters–patients can be missing, which further defines a complete vs. incomplete study design.

The general framework used is a two-way Analysis Of Variance (ANOVA), where the total variability can be decomposed between the patients’ (p) measurement difference, operators’ (r) difference, and residual random error. In this specific context, we can, for example, determine that a specific measurement *M_m_* be written as follows in Equation (5):(5)$$M_{rp}=\mu +\mu_{r}+\mu_{p}+\mu_{rp}$$
where *μ* is the mean of *M_m_* in the tested population; *μ_r_* is the specific quantity of the operators/technicians; and *μ_p_* is the specific error term due to the patient’s interaction with the operator/technician—the veterinary patient effect (*μ_rp_*)—which also includes a random part since the veterinary patients are only measured once per operator/technician. The variance of the measure can, therefore, be partitioned as in Equation (6).
(6)$$\sigma_{M_{rp}}^{2}=\sigma_{r}^{2}+\sigma_{p}^{2}+\sigma_{rp}^{2}$$

In the one-way design, the patient is nested within one specific operator, so the effect of the operator cannot be distinguished from the patient, known as veterinary–patient error. (Equation (6) is simplified by removing the $\sigma_{r}^{2}$ term, which is confounded in the operator’s veterinary–patient error). This general framework is then used for defining different types of ICC based on the partition of veterinary–patient variance vs. veterinary–patient plus error variance (Table 1). The ICC can be differentiated based on (1) agreement vs. consistency, including or not the variance of the operator effect ($\sigma_{r}^{2}$); (2) average vs. single ratings, where the operator-related variance is divided by the number of operators (k) per patient; and (3) random vs. fixed operators, where a specific part-variance ($\sigma_{pr-\varepsilon }^{2}$) is subtracted from the veterinary–patient variance ($\sigma_{p}^{2}$) in the numerator. For this reason, fixed-operator ICC can only be estimated if the operators are measuring the same veterinary patient multiple times or if the veterinary patient via the operator–interaction effect is assumed to be absent. One can easily see from Table 1 that for a specific ICC, agreement ICC is generally lower than the consistency ICC and that the random-effect ICC is lower than the fixed-effect ICC. No fixed-effect ICC can be established for one-way designs in the absence of a distinction between the variance of the operator and the variance of the veterinary patient, known as the operator interaction, which is confounded. Several considerations should be taken into account for the selection of the specific ICC, to report as reviewed by Ten Hove et al. [8]. They proposed a flow chart that helps scientists select which ICC to report depending on the study design and aim of the reliability assessment. Basically, consistency examines whether the operators or technicians are classifying the same subjects with low and high values, even if an absolute difference score is present. This means that the ranking of the measurements obtained by the different operators or technicians is comparable, despite some absolute differences in scoring being observed. The absolute agreement is more interested in assessing how the values given to a specific veterinary patient by different raters are close and not the relative ranking of patients’ values per se. In short, we are interested in consistency when we are not interested in the systematic difference between operators/technicians (absolute agreement). Both indicators (absolute agreement and consistency ICC) can be useful depending on the specific context of the intended application of the measurement under investigation. The choice between the random and fixed model in two-way models is associated with the selection of the operators/technicians. If the operators/technicians are randomly selected from a population of operators/technicians, or if there is an extrapolation to operators/technicians with the same characteristics as those used in the study, a random model is preferred. When focusing only on the specific operators/technicians used for the study, a fixed-rater effect can be preferred.

The choice of reporting one or multiple ICCs depends on the type of study and the questions the authors are trying to answer [8]. Unfortunately, the type of ICC calculated and reported is uncommonly described in the medical literature [9]. Some ICCs also have other specific names. For example, ICC(C,k) as the average measure, consistency ICC, is more commonly known as Cronbach’s alpha (α). This α value is commonly used in psychometric tests where different questions are supposed to assess the same specific concepts or construct. A high Cronbach’s α generally indicates that these questions are measuring the same concept or construct. A specific discussion of Cronbach’s alpha is outside the scope of this manuscript. The reader is referred to other references on that specific reliability parameter [1,10].

## 5. Comparing Two Different Laboratory Measurements (e.g., Metabolite M Measured Using Two Different Devices or Measured Repeatedly with the Same Device)

The previous ICC approach can be extended to various contexts, especially when intra- and inter-operator reliability is required. However, when focusing on a comparison between two different techniques to quantitatively assess a specific marker, as commonly encountered in laboratory analyses, specific statistical analyses have been mentioned that can elucidate to what extent a specific technique can be compared to another.

### 5.1. Lin’s Concordance Correlation Coefficient

When comparing two closely related measurements (M_2_ (mean *μ*_2_; variance $\sigma_{2}^{2}$) vs. M_1_ (*μ*_1_;$\;\sigma_{1}^{2}$) as a gold standard), the concordance correlation coefficient (CCC) has been defined as a way to estimate the perpendicular squared deviation from the forty-five-degree (y = x) line [11]. The specific calculation of the CCC also accounts for the covariance between M_1_ and M_2_ with Cov(M_1_,M_2_) = σ_12_, as presented in Equation (7):(7)$$CCC=\frac{\sigma_{12}}{{\left(\mu_{1}-\mu_{2}\right)}^{2}+\sigma_{1}^{2}+\sigma_{2}^{2}}$$

It can also be demonstrated that the CCC can be further decomposed as a term proportional to Pearson’s R correlation (Equation (8)):(8)$$CCC=R*C=R*\frac{2}{v+\frac{1}{v}+u^{2}}$$
where $v=\frac{\sigma_{1}}{\sigma_{2}}$; $u=\frac{{\left(\mu_{1}-\mu_{2}\right)}^{2}}{\sigma_{1}\sigma_{2}}\;$; v is a scale shift (how far the corresponding slope differs from the 45° line); and u is a location shift (as the intercept different from 0). The CCC is, therefore, an interesting way to look for the reproducibility of a specific measurement; the closer it is to 1, the better the reproducibility of the measurement. Specific benchmarks have been reported for helping its clinical interpretation with poor (CCC < 0.90), moderate (between 0.90 and <0.95), substantial (between 0.95 and <0.99), and almost-perfect when higher than 0.99 [12]; however, similar to correlation coefficients R and r, these are empirical benchmarks. In the dataset reported in Figure 1, the CCC is 0.794 (see Appendix A).

### 5.2. Determining the Coefficient of Variation (CV) of a Repeated Measurement

Any measurement comes with a specific error due to the measurement technique. It is of utmost importance to characterize this type of error, in order to know if the new measurement method has a variability and if this variability depends on specific values of the quantity of interest. The coefficient of variation is simply the standard deviation of the two measurements divided by their mean (CV = $\frac{\sigma }{\mu }$).

When developing the calculation of the CV for k different samples (${m1}_{1,\ldots ,k}$, ${m2}_{1,\ldots ,k}$), the CV can be written in a function of the repeated-pair differences ($m_{1i}-m_{2i}$) and means ($m_{1i}+m_{2i})/2$), as presented in Equation (9):(9)$$CV=\sqrt{\frac{\sum_{i=1}^{k}\frac{{\left(m_{1i}-m_{2i}\right)}^{2}/2}{{\left(\left(m_{1i}+m_{2i}\right)/2\right)}^{2}}}{k}}$$

The CV is commonly reported in clinical chemistry as a way to quantify test–retest reliability; however, it has been largely criticized in recent years as the standard deviation may naturally increase with the measurement mean [13]. This non-proportionality can be an important problem, especially when data are not normally (e.g., log-normally) distributed. Moreover, the CV’s calculation is compromised with measurements with a null mean. In a recent review of the CV’s limitations by Pélabon et al. [13], the authors specifically mentioned not using the CV for nominal, ordinal, interval, or different variables.

### 5.3. Exploring Proportional and Differential Bias Using Robust Approaches

There are different types of errors between two measurements of the same marker with two different devices. The differences are generally described as a constant error term and a proportional error term. This can simply be summarized in Equation (10):(10)$$M_{Analyzer\_new}=\alpha +\beta *M_{Analyzer\_ref}+\varepsilon$$
where *α* is the constant or systematic bias term; and *β* is the proportional bias. In the case of *β* = 1, only a constant bias is present. Robust approaches to compare these two measurements are the Passing–Bablok [14] and Deming regressions [15]. These approaches are particularly helpful when data are not normally distributed or heteroskedastic (e.g., an increase in variance proportional to the value to be measured), which is frequently encountered in many different clinical situations.

#### 5.3.1. Deming Regression

The Deming regression is an extension of linear regression that also accounts for the error of the new and current method (i.e., assuming not only the new method measurement $M_{New\_method}$ has an error (${M_{New\_method}}_{i}={{M_{New\_method}}^{*}}_{i}+{\varepsilon_{new}}_{i}$) but also that the comparator method ($M_{Current\_method}$) has inherent measurement error (${M_{Current\_method}}_{i}={{M_{Current\_method}}^{*}}_{i}+{\varepsilon_{current}}_{i}$), as represented in Figure 2. The Deming regression assumes that the errors ($e_{new},\;e_{current}$) of both measures are independent and distributed normally. An important assumption is also that the ratio of variance $\frac{\varepsilon_{new}^{2}}{\varepsilon_{current}^{2}}$ is constant. In contrast to ordinary linear regression, which minimizes the sum of distances between the Y values and the fitted line, the Deming regression minimizes the distances in both axes (X,Y) directions. When the data seems largely heteroskedastic (e.g., a proportional increase in variance ratio), a weighted Deming regression can also be used to allocate specific weights to the data points (i.e., reciprocal of the squared reference value). The conditions of application for Deming or weighted Deming regressions should be assessed and tested when relevant [16].

#### 5.3.2. Passing–Bablok Regression

Despite the fact that the Deming regression approach has less restrictive assumptions than the linear model, it still relies on assumptions of the constant or the proportional variance of both measurements. When these assumptions do not hold, the Passing–Bablok approach could be used due to its robustness. The objective of the Passing–Bablok regression is roughly to determine if α and β in Equation (10) are different from 0 and 1, respectively, based on 95% CI, including these values or not. A specific assumption is that the relationship between the two measurements is linear, as generally verified by a cumulative sum control chart (CUSUM) test. This method also assumes that measurement errors in both methods have the same distribution (which is not necessarily normal) and a constant ratio of variance. The estimation of β is based on the shifted median of all slopes, formed by possible data point pairs (shifted only means that the numbers of pairs with slope < −1 are accounted for correcting the median). This approach is considered robust to various data distributions and error distributions [14]. The Passing–Bablok estimates are robust to outliers (Figure 2) and can be used in various contexts where Deming regression assumptions are not satisfied.

### 5.4. Agreement (Bland–Altman) Plot

The relationship between two closely related continuous variables (e.g., two measurements of the same metabolite using a reference analyzer and a new one, or measurement of a specific measurement using the same ultrasound unit by two different raters M*_m1_* and M*_m2_*, respectively) can be further evaluated using a specific approach firstly described by Bland and Altman in their seminal article [17]. This analysis quantifies the agreement by defining the limits of agreements, mean, and standard deviation of the bias. This approach has been extensively used in medicine because it is visually and clinically intuitive [18]. The agreement plot indicates the difference between the two measurements (*Y*-axis: (*M_m_*_1_–*M_m_*_2_) vs. the mean of the two measurements (*X*-axis: (*M_m_*_1_ + *M_m_*_2_)/2). The difference should lie between +/− 2 standard deviations of the mean difference (upper and lower limits of agreement). The graphical appearance of the Bland–Altman analysis contains, therefore, three different lines and their associated confidence intervals (mean bias, and the upper and lower limits of agreements), as presented in Figure 3. It can be easily seen if the cloud of dots is homogenously spread around the horizontal mean bias line or if the dots’ repartition differs when the mean measurement increases. In the latter case, the definition of a mean bias is not meaningful. It is also important to consider that this approach is not meaningful for ordinal scores because of the absence of clinical meaning of both the mean bias and limits of agreement. The ordinal scoring preferred for assessing reliability is to determine the ICC as previously emphasized.

The Bland–Altman approach estimates the average bias and constant limits of agreement (i.e., three parallel lines); however, this calculation is based on important assumptions that have been recently reviewed by Taffé [18]. The first assumption is that the bias is constant across the measurement ranges since the “average” bias is calculated. Then, the errors are also assumed to be constant across the measurement ranges, which needs to be consistently verified. Finally, the measurement error variances are supposed to be the same for both methods. For these reasons, it is important to know these limitations to put in perspective the potential applications of the Bland–Altman plot. The agreement plot was further extended, accounting for non-constant bias; therefore, allowing proportional and differential bias, which were further obtained from the slope (β) and intercept (α) of the bias regression in Equation (11) (Figure 4).
(11)$$M_{m1}-M_{m2}=\alpha +\beta *\left[\frac{\left(M_{m1}+M_{m2}\right)}{2}\right]+error$$

The differential and proportional biases are then defined in Equations (12) and (13):(12)$$\mathrm{Differential}\;\mathrm{bias}=\frac{2*\alpha }{2-\beta }$$
(13)$$\mathrm{Proportional}\;\mathrm{bias}=-\frac{2+\beta }{\beta -2}$$

Variation in the limits of agreement was also allowed with non-parallel limits-of-agreement lines.

Despite the improvement of exploration of the variability in differences, a limitation of the traditional Bland–Altman analysis is that it does not allow for exploring each measurement separately. Most of the time, as previously reported in the measurement theory, none of the two measurements can be considered as truly assessing the trait under investigation. Both *M_m_*_1_ and *M_m_*_2_ are trying to assess the true *M* value with specific errors. The work from Taffé extends on Bland and Altman’s since the mean of (*M_m_*_1_ + *M_m_*_2_)/2 used by default in the Bland–Altman analysis is not an unbiased estimate of M [19]. In his seminal work, Taffé proposed an empirical Bayesian method to compute the best linear unbiased prediction (BLUP) of M [20]. In other cases, using one of the measures, *M_m_*_1_ or *M_m_*_2_ values, as an unbiased estimate of M only works if one of the two measurements can be considered as a perfect reference standard test, which is not often the case.

Basically, the approach from Taffé extends the differential (*α*_1_, *α*_2_) and proportional (*β*_1_, *β*_2_) bias of *M_m_*_1_ and *M_m_*_2_ to determine the true *M* value [Equations (14) and (15)].
(14)$$M_{m1}=\alpha_{1}+\beta_{1}*M+\varepsilon_{1}$$
(15)$$M_{m2}=\alpha_{2}+\beta_{2}*M+\varepsilon_{2}$$

We previously assumed that the specific measurement error was normally distributed around a 0 mean and constant variance ($\sigma_{\varepsilon }^{2}$); however, the constant variance assumption can be relaxed, allowing variation with the specific measure under interest ($\sigma_{M}^{2}$), therefore, indicating that the variance depends on the quantity of *M*. Briefly, the lower (or higher, respectively) *M* is, the lower (or higher, respectively) its variance is expected to be.

Finally, when one of the two methods does not have any measurement error, the BA method should not be performed as recently demonstrated [19]. In this case, the BA method will produce biased estimates of the difference between methods. A simple linear regression of the error between the second method over the method of reference could then be performed.

## 6. Discussion

As I have shown in the current review, reproducibility, agreement, and reliability are complex concepts that can be assessed using various methods that are complementary and depend on the objective of the researchers, the variable to assess, the study design, as well as the prior definition of what is acceptable from a clinical or a research perspective. Despite the fact that several benchmarks have been reported, they are empirical and the researchers should define what is acceptable depending on the variable measured. Most of the parameters and tests available to determine reliability are also based on several assumptions that need to be known to select the appropriate method. I did not address the specific issue of sample size determination to test an a priori hypothesis of reliability; however, this is of utmost importance to perform sample size determination before planning a study, to be able to interpret correctly the results from agreement and reliability analyses [21]. Knowing to what extent the variability in a measurement is associated with the real variation in the trait to measure vs. due to measurement error is of utmost importance for being able to interpret correctly the observed values of the parameter of interest.

I propose a general framework to help the veterinary practitioner or veterinary researcher cope with these complex concepts depending on the type variable they are trying to assess in Figure 5. Despite our primary focus being on the reliability assessment of quantitative measurements, ordinal measurements such as scoring systems are also commonly used in veterinary medicine and their reliability assessment depends on the use of ICC [1,2,3]. Using other methods such as agreement plots is not recommended because establishing a mean bias cannot be interpreted in ordinal scales.

## 7. Conclusions

As I have shown in the current review, agreement and reliability are complex concepts that can be assessed using various methods that are complementary and depend on the measurement of interest, the objective of the researchers, as well as the prior definition of what is acceptable from a clinical or a research perspective. Most of the parameters and tests available are also based on several assumptions that need to be known to select the appropriate method. I did not address the specific issue of sample size determination to test an a priori hypothesis of reliability; however, this is of utmost importance to perform sample size determination before planning a study to be able to interpret correctly the results from agreement and reliability analysis. Knowing to what extent the variability in a measurement is associated with the real variation in the trait to measure vs. due to measurement error is of utmost importance for being able to interpret correctly the observed values of the parameters of interest.

## Figures and Tables

**Figure 1 animals-13-02793-f001:**
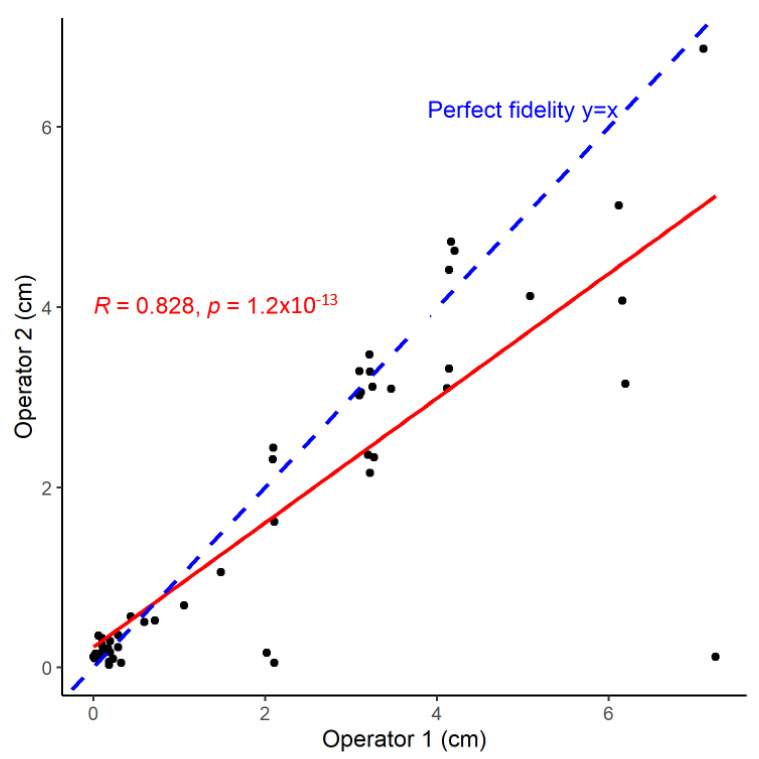
Correlation between two measurements obtained by two different operators. The linear regression of operator 2 (in cm) over operator 1 is indicated as a red line. The perfect identity line is indicated as a blue dashed line. The *p*-value is the level of confidence in the null hypothesis (R = 0, no correlation between the 2 measurements), meaning we can safely reject the null hypothesis. However, correlations do not assess reliability between the 2 measurements. Spearman’s r is a more natural choice in this case, as it is more robust to various types of data distribution (i.e., deviating from linear regression assumptions). In this case r = 0.773, *p* < 2.2 × 10^−16^.

**Figure 2 animals-13-02793-f002:**
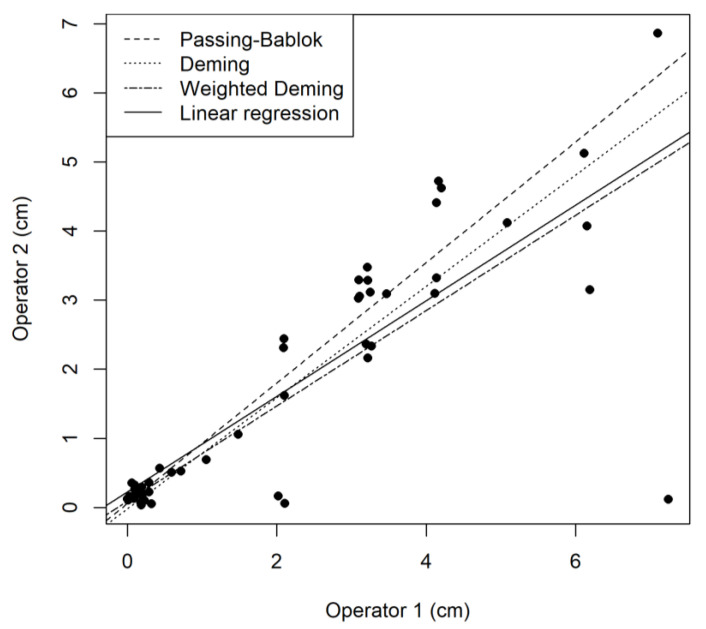
Different indicators of linear relationship between 2 measurement methods of the same variable. Linear (full line), Passing–Bablok (dashed line), Deming (dotted line), and weighted Deming regression (two-dashed line) lines are indicated.

**Figure 3 animals-13-02793-f003:**
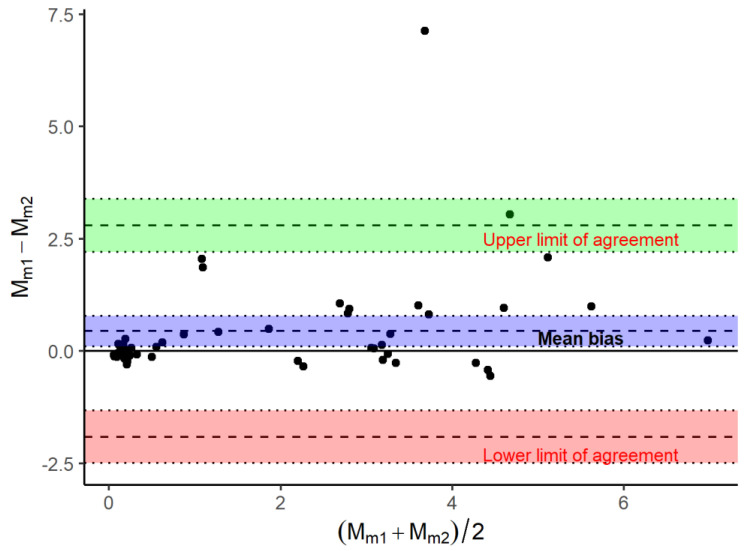
Agreement (Bland–Altman) plot. This figure summarizes the difference between both measurements (*M_m_*_1_–*M_m_*_2_) as a function of the mean measurement (in the absence of one of the techniques, *M_m_*_1_ or *M_m_*_2_, to be considered as a gold standard test). The mean bias and its associated 95% CI are presented in blue. The lower and upper limits of agreement are also highlighted, as well as their associated 95% confidence intervals (red and green, respectively).

**Figure 4 animals-13-02793-f004:**
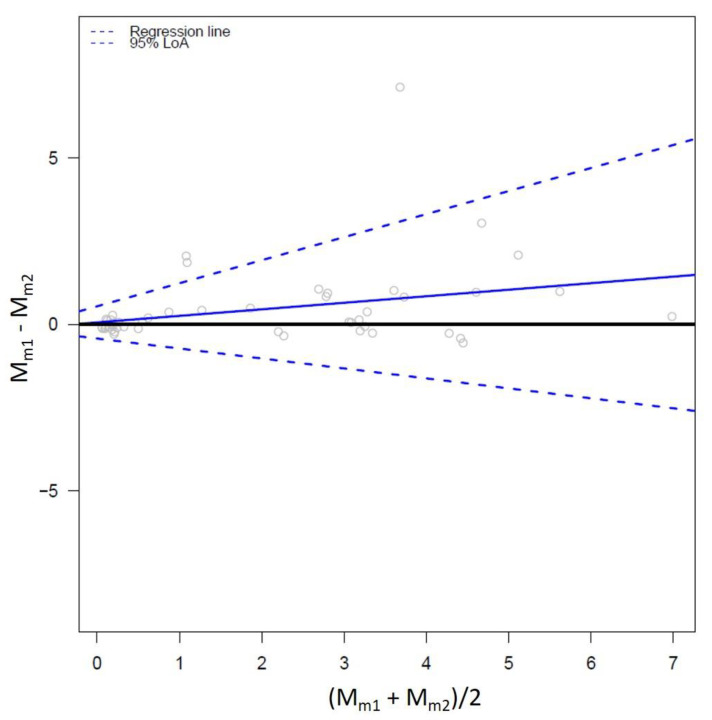
Adjusted agreement plot allowing differential and proportional bias (blue line) and associated proportional limits of agreement lines (upper and lower dashed blue lines). The grey circles represent the data points.

**Figure 5 animals-13-02793-f005:**
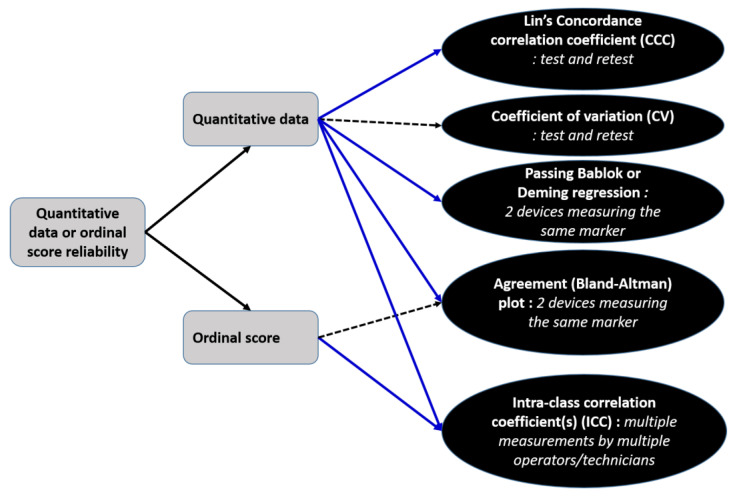
Proposed framework for deciding which reliability indicator to use when trying to assess quantitative or ordinal measurement in veterinary medicine. The blue arrows indicate the natural choices whereas the dotted arrows indicate suboptimal choices due to limitations of the indicators vs. intended use.

**Table 1 animals-13-02793-t001:** Intra-class coefficient correlation determination based on the partition of variance associated with patients (*p*), *k* operators/technicians, and (*r*) sources of variance.

Design	Type of ICC ^1^	Random vs. Fixed	Intra-Class Coefficient	
			Single Ratings	Average Ratings
2-way	Absolute (A)	Random	$ICC\left(A,1\right)=\frac{\sigma_{p}^{2}}{\sigma_{p}^{2}+\sigma_{r}^{2}+\sigma_{pr}^{2}\;}$	$ICC\left(A,\;k\right)=\frac{\sigma_{p}^{2}}{\sigma_{p}^{2}+(\sigma_{r}^{2}+\sigma_{pr}^{2})/k\;}$
		Fixed	$ICC\left(A,1\right)=\frac{\sigma_{p}^{2}-\sigma_{pr-\varepsilon }^{2}/\left(k-1\right)}{\sigma_{p}^{2}+\sigma_{r}^{2}+(\sigma_{pr-\varepsilon }^{2}+\sigma_{\varepsilon }^{2})\;}$	$ICC\left(A,k\right)=\frac{\sigma_{p}^{2}-\sigma_{pr-\varepsilon }^{2}/\left(k-1\right)}{\sigma_{p}^{2}+(\theta_{r}^{2}+(\sigma_{pr-\varepsilon }^{2}+\sigma_{\varepsilon }^{2})/k\;}$
	Consistency (C)	Random	$ICC\left(C,1\right)=\frac{\sigma_{p}^{2}}{\sigma_{p}^{2}+\sigma_{pr}^{2}\;}$	$ICC\left(C,k\right)=\frac{\sigma_{p}^{2}}{\sigma_{p}^{2}+\sigma_{pr}^{2}/k\;}$
		Fixed	$ICC\left(C,1\right)=\frac{\sigma_{p}^{2}-\sigma_{pr-\varepsilon }^{2}/\left(k-1\right)}{\sigma_{p}^{2}+(\sigma_{pr-\varepsilon }^{2}+\sigma_{\varepsilon }^{2})\;}$	$ICC\left(C,1\right)=\frac{\sigma_{p}^{2}-\sigma_{pr-\varepsilon }^{2}/\left(k-1\right)}{\sigma_{p}^{2}+\sigma_{pr}^{2}/k\;}$
1-way	______	Random	$ICC\left(1\right)=\frac{\sigma_{p}^{2}}{\sigma_{p}^{2}+\sigma_{pr}^{2}\;}$	$ICC\left(k\right)=\frac{\sigma_{p}^{2}}{\sigma_{p}^{2}+\sigma_{pr}^{2}/k\;}$

^1^ ICC: intra-class correlation coefficient. Depending on the variance partitioning and study design, several types of ICC are distinguished.

## Data Availability

The data were derived from the open-access publication [3]. They can freely be downloaded using the following link (https://onlinelibrary.wiley.com/action/downloadSupplement?doi=10.1111%2Fjvim.15257&file=jvim15257-sup002-suppinfo02.pdf, accessed on 30 August 2023).

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
