# Peer review of "Reliability Associated with the Measurement of Continuous Variables in Veterinary Medicine: What the Different Possible Indicators Tell, and How to Use and Report Them"

_animals, 2023, doi:10.3390/ani13172793_

Round 1
Reviewer 1 Report
In the present article, the author has revised the multiple indicators that can be utilized to determine the sources of variability to attain robust indicators useful when trying to quantify test-retest reliability. The following are some minor comments to be considered:
1. The English language writing needs to be revised well and improved : Line 9; Veterinary science in based on, do you mean is based on?
Line 12; we review the multiple indicators, since the present article is authored by one author only, it is better to use I instead of we.
Line 55; confusion for the researcher trying to assess. use researchers instead of researcher.
Line 135; quantitative variables and is a commonly. Remove "and".
2. The present article focus on the veterinary medicine, so that it is better to use case instead of patient.
3. Line 168 "The variable values per se are not used for". what se refers to?
4. Line 170; "The higher is its value the higher is the correlation between the ranked variables". This sentence needs to be rephrased in an easier and proper way.
The English language needs to be checked, improved, and represented in an easy to understand way. Several grammatical errors are present and needs to be corrected.
Author Response
In the present article, the author has revised the multiple indicators that can be utilized to determine the sources of variability to attain robust indicators useful when trying to quantify test-retest reliability. The following are some minor comments to be considered:
Authors (AU): I thank the reviewer for his/her critical review of the manuscript. I have tried to address every comment or concerns of the reviewer in the current report. I sincerely hope the manuscript is now acceptable for publication.
1. The English language writing needs to be revised well and improved : Line 9; Veterinary science in based on, do you mean is based on?
AU: I am sorry for this mistake. This has been changed.
Line 12; we review the multiple indicators, since the present article is authored by one author only, it is better to use I instead of we.
AU: this has been changed throughout the text as suggested.
Line 55; confusion for the researcher trying to assess. use researchers instead of researcher.
AU: changed as suggested
Line 135; quantitative variables and is a commonly. Remove "and".
AU: sorry for this mistake. This has been changed.
- The present article focus on the veterinary medicine, so that it is better to use case instead of patient.
AU: sorry for the confusion. Based on further reading to understand the difference between cases and patients (https://files-aje-com.s3.amazonaws.com/www/row/_assets/docs/AJE-Clinical-Terminology-and-Phrasing-Resource-2015.pdf ) we can notice that patient has a broader meaning than case which is traditionally associated with a specific condition. The context of application of this review is larger than application to sick animals (e.g. the clinician can measure body weight with a scale or a measuring tape in a group of animals with or without disease). For this reason and to avoid any confusion we have used “veterinary patient” throughout the text.
- Line 168 "The variable values per se are not used for". what se refers to?
AU: The term “per se” is the Latin adverbial formula meaning “by itself” or “of itself”. We have used “by themselves” in the revised version of the manuscript to improve text readability.
- Line 170; "The higher is its value the higher is the correlation between the ranked variables". This sentence needs to be rephrased in an easier and proper way.
AU: I have modified the sentence as requested.
Comments on the Quality of English Language
The English language needs to be checked, improved, and represented in an easy to understand way. Several grammatical errors are present and needs to be corrected.
AU: I sincerely hope that the English writing is better now.

Reviewer 2 Report
I enjoyed reading this well written manuscript that addresses an area that is seldom discussed in veterinary medicine. I only have some minor comments/suggestions (see below).
General comments:
-There is no statement regarding ethical approval for collection of or use of the data presented in this manuscript. The original study where these data were collected should be cited in the main manuscript as well as the supplementary material and a statement that the data are presented with permission and in accordance with the terms of the ethical approval of the original study should be included. I would suggest including this at the end of the introduction as a short explanation, although this could also go at the end after the conclusion.
-There are no acknowledgements in this manuscript which I find to be unusual given the collaborative nature of research in general and the fact that real data were used for illustration in this manuscript. I note that the original work has 6 authors so please just double check that you have not neglected to thank other members of the group who might have contributed to this work (e.g. by permission of data re-use etc.).
-Several times the abbreviation 'ex' is used. This is not a common abbreviation in English and it is unclear what is meant by this. If this is intended to convey 'example' I would suggest replacing 'ex' with the more standard 'e.g.'. If, however you wish to continue using 'ex' it would be helpful to the reader for this to be written in full the 1st time it is used
-Throughout the plural we/our are used for the active voice; however, as this is a single author paper I/my would be more grammatically correct
-Figures 2 and 3 contain red and green which will be difficult for colour-blind people to read. Consider changing the colours to a more accessible palette. This particularly applies to Figure 2 because you refer to different lines by their colour only. Figure 3 has the upper and lower limit labelled so this would still be readable even if you did not know which band was which colour.
Specific comments:
Line 10: I find the phrase '...variability of the variable to measure itself...' a bit confusing. Consider rephrasing as '...variability of the variable measured.' for improved clarity
Line 19-20: The phrase 'Reliability of a measurement assess the variability of the measurement....' is quite confusing and I am not completely clear what you mean here. Should this state 'assesses the variability'? If so, please correct the typo but otherwise consider rephrasing the sentence for improved clarity
Lines 58-60: The statement in these lines ('A test or measurement is said to be reliable...') would benefit from a supporting reference
Line 382: The statement that BA plots are favoured because they are visually appealing would benefit from a reference to support it. I think without a reference, you cannot make this assertion and if there are no data to support this statement I would suggest rephrasing to be less dogmatic, or remove the reference to visual appeal altogether.
Figure 1 and lines 133-180: in this section Figure 1 is referred to with respect to Spearman rank correlation. Only a very small part of Figure 1 refers to the Spearman rank correlation and it is not a very visual aspect of Figure 1. I found this a little confusing and I would consider clarifying what you are referring to here, or highlighting the Rho and p-values a bit more, so it is easier for the reader to see what is being referred to.
Figure 2: The Figure legend does not quite match the figure itself - the red line in the graph is dashed but it is solid in the legend. Please correct this.
Figure 4: As you only use one shade of blue in this figure it is unnecessary to refer to 'dark blue' ('blue' is sufficient)
Figure 5: The dashed arrows are very difficult to see - consider using a darker colour.
Author Response
I enjoyed reading this well written manuscript that addresses an area that is seldom discussed in veterinary medicine. I only have some minor comments/suggestions (see below).
Authors (AU): I thank the reviewer for his/her critical review of the manuscript and the general positive comments. I have tried to address the specific concerns and comments in the revised version of the manuscript.
General comments:
-There is no statement regarding ethical approval for collection of or use of the data presented in this manuscript. The original study where these data were collected should be cited in the main manuscript as well as the supplementary material and a statement that the data are presented with permission and in accordance with the terms of the ethical approval of the original study should be included. I would suggest including this at the end of the introduction as a short explanation, although this could also go at the end after the conclusion.
AU: I really thank the author for this specific comment. Ihave now tried to improve this aspect by clearly adding the data taken for illustrating the different figures of the manuscript. I sincerely hope that this is clearer now.
-There are no acknowledgements in this manuscript which I find to be unusual given the collaborative nature of research in general and the fact that real data were used for illustration in this manuscript. I note that the original work has 6 authors so please just double check that you have not neglected to thank other members of the group who might have contributed to this work (e.g. by permission of data re-use etc.).
AU: I totally agree on the fact that research is collaborative by definition. I have used the data from a previous open access publication which on the reliability of ultrasonographic lung consolidation by different operators. This publication also provides open access raw data that are available online (https://onlinelibrary.wiley.com/action/downloadSupplement?doi=10.1111%2Fjvim.15257&file=jvim15257-sup002-suppinfo02.pdf ). I have clarified this in the acknowledgement section as well as in the definition section. I have also cited the reference from where the data were extracted as suggested.
-Several times the abbreviation 'ex' is used. This is not a common abbreviation in English and it is unclear what is meant by this. If this is intended to convey 'example' I would suggest replacing 'ex' with the more standard 'e.g.'. If, however you wish to continue using 'ex' it would be helpful to the reader for this to be written in full the 1st time it is used
AU: I am sorry for this confusion. The reviewer is right that e.g. would be more suitable. I have clarified this throughout the manuscript.
-Throughout the plural we/our are used for the active voice; however, as this is a single author paper I/my would be more grammatically correct
AU: This has been changed as requested.
-Figures 2 and 3 contain red and green which will be difficult for colour-blind people to read. Consider changing the colours to a more accessible palette. This particularly applies to Figure 2 because you refer to different lines by their colour only. Figure 3 has the upper and lower limit labelled so this would still be readable even if you did not know which band was which colour.
AU: Thank you for notice this. We have modified these figures accordingly.
Specific comments:
Line 10: I find the phrase '...variability of the variable to measure itself...' a bit confusing. Consider rephrasing as '...variability of the variable measured.' for improved clarity
AU: Changed as suggested.
Line 19-20: The phrase 'Reliability of a measurement assess the variability of the measurement....' is quite confusing and I am not completely clear what you mean here. Should this state 'assesses the variability'? If so, please correct the typo but otherwise consider rephrasing the sentence for improved clarity
AU: Changed as suggested.
Lines 58-60: The statement in these lines ('A test or measurement is said to be reliable...') would benefit from a supporting reference
AU: Reference added as suggested.
Line 382: The statement that BA plots are favoured because they are visually appealing would benefit from a reference to support it. I think without a reference, you cannot make this assertion and if there are no data to support this statement I would suggest rephrasing to be less dogmatic, or remove the reference to visual appeal altogether.
AU: We’ve modified this sentence according to the reviewer concern.
Figure 1 and lines 133-180: in this section Figure 1 is referred to with respect to Spearman rank correlation. Only a very small part of Figure 1 refers to the Spearman rank correlation and it is not a very visual aspect of Figure 1. I found this a little confusing and I would consider clarifying what you are referring to here, or highlighting the Rho and p-values a bit more, so it is easier for the reader to see what is being referred to.
AU: We’ve changed the figure information focussing more on the Pearson R and indicating Spearman rho in the text. Unfortunately, it is very difficult to get a visualisation of Spearman rho because it is a rank-based indicator.
Figure 2: The Figure legend does not quite match the figure itself - the red line in the graph is dashed but it is solid in the legend. Please correct this.
AU: We are very sorry for that mistake. We have redrawn this figure to improve its readability and not relying only color description.
Figure 4: As you only use one shade of blue in this figure it is unnecessary to refer to 'dark blue' ('blue' is sufficient)
AU: modified as suggested.
Figure 5: The dashed arrows are very difficult to see - consider using a darker colour.
AU: changed as suggested.

Reviewer 3 Report
The manuscript reviews the statistical methods used to determine reliability of measurements of continuous variables.
Although, it does a nice review of statistical methods, it fails to present practical examples from the literature.
I recommend the authors to review the manuscript by Lean et al 2016 Invited review: Recommendations for reporting intervention studies on reproductive performance in dairy cattle: Improving design, analysis, and interpretation of research on reproduction
Minor details
line 12 - replace article by manuscript
line 25 - replace article by manuscript
line 56 - replace aim by objective
line 56 - replace article by manuscript
throughtout the manuscript replace rates by technician
The English is appropriate
Author Response
The manuscript reviews the statistical methods used to determine reliability of measurements of continuous variables.
Although, it does a nice review of statistical methods, it fails to present practical examples from the literature.
I recommend the authors to review the manuscript by Lean et al 2016 Invited review: Recommendations for reporting intervention studies on reproductive performance in dairy cattle: Improving design, analysis, and interpretation of research on reproduction
AU: I thank the reviewer for the time spent to the review of our manuscript as well as this comment. Thank you for this suggestion. I have more explicitly described the data that were used. I think that one of the main limitations in the current veterinary literature is that it is difficult to find dataset with individual values of the different measurements (repeated at either individual or operator level). This is why it is challenging to find many different practical examples. We have tried to answer to all comments and questions raised by the reviewer and sincerely hope that the manuscript is now acceptable for publication.
Minor details
line 12 - replace article by manuscript
AU: Replaced as suggested.
line 25 - replace article by manuscript
AU: Replaced as suggested.
line 56 - replace aim by objective
AU: Replaced as suggested.
line 56 - replace article by manuscript
AU: Replaced as suggested.
throughtout the manuscript replace rates by technician
AU: Replaced as suggested.
The English is appropriate
AU: Thank you.
